# Effect of Parylene C on the Corrosion Resistance of Bioresorbable Cardiovascular Stents Made of Magnesium Alloy ‘Original ZM10’

**DOI:** 10.3390/ma15093132

**Published:** 2022-04-26

**Authors:** Makoto Sasaki, Wei Xu, Yuki Koga, Yuki Okazawa, Akira Wada, Ichiro Shimizu, Takuro Niidome

**Affiliations:** 1Faculty of Advanced Science and Technology, Kumamoto University, Kumamoto 860-8555, Japan; niidome@kumamoto-u.ac.jp; 2Japan Medical Device Technology, Co., Ltd., Kumamoto 861-2202, Japan; y.koga@jmdt.co.jp (Y.K.); a.wada@jmdt.co.jp (A.W.); 3Fuji Light Metal, Co., Ltd., Kumamoto 869-0192, Japan; yuuki-okazawa@fujisash.net; 4Department of Mechanical Engineering, Okayama University of Science, Okayama 700-0005, Japan; shimizu@mech.ous.ac.jp

**Keywords:** corrosion resistance, magnesium alloy, bioresorbable stent, parylene C, surface coating

## Abstract

Magnesium (Mg) alloy has attracted significant attention as a bioresorbable scaffold for use as a next-generation stent because of its mechanical properties and biocompatibility. However, Mg alloy quickly degrades in the physiological environment. In this study, we investigated whether applying a parylene C coating can improve the corrosion resistance of a Mg alloy stent, which is made of ‘Original ZM10’, free of aluminum and rare earth elements. The coating exhibited a smooth surface with no large cracks, even after balloon expansion of the stent, and improved the corrosion resistance of the stent in cell culture medium. In particular, the parylene C coating of a hydrofluoric acid-treated Mg alloy stent led to excellent corrosion resistance. In addition, the parylene C coating did not affect a polymer layer consisting of poly(ε-caprolactone) and poly(D,L-lactic acid) applied as an additional coating for the drug release to suppress restenosis. Parylene C is a promising surface coating for bioresorbable Mg alloy stents for clinical applications.

## 1. Introduction

Coronary vascular disease is a major cause of death in the developed world [1]. Percutaneous coronary intervention combining balloon angioplasty and stent implantation is a primary treatment for coronary heart disease [2]. Drug-eluting stents are now widely used because they can reduce in-stent restenosis and target lesion revascularization compared with bare-metal stents [3]. Sirolimus (SRL) is an antiproliferative drug widely used in drug-eluting stent applications [4]. However, drug-eluting stents have been reported to impair endothelial regeneration, leading to an increased risk of very late stent thrombosis in patients after stent implantation [5]. Thus, bioresorbable stents have been developed to address the problems associated with conventional metallic stents [6]. The bioresorbable stents made of biodegradable polymers or bioresorbable metals have become major trends. Compared with biodegradable polymer stents, bioresorbable metals such as magnesium, zinc, and iron have better mechanical properties and are expected as new-generation stent materials [7].

Magnesium (Mg) alloy is expected as the new generation metal for use in the development of bioresorbable stents because it exhibits good mechanical properties and its corrosion products are non-toxic [8]. However, Mg alloy exhibits rapid corrosion in the physiological environment, limiting its successful clinical application [9]. A vascular stent is required to maintain its mechanical strength for at least 6 months and to be resorbed over 6 months to 2 years [10,11]. Therefore, enhancing the anti-corrosion properties of Mg alloy is key to its clinical application as a bioresorbable stent.

To improve the corrosion resistance of Mg alloy, the addition of rare earth element to Mg alloy [12,13], surface coating [14,15], and chemical conversion treatment [16,17] have been widely studied. The rare earth elements refer to the metals yttrium (Y), dysprosium (Dy), neodymium (Nd), gadolinium (Gd), etc. However, the suitability and biocompatibility of rare earth elements are still considered carefully [18]. Conventional Mg alloy, i.e., WE43, contains rare earth elements and is suspected to show cytotoxicity [19]. Moreover, AZ31B contains aluminum, which is not a rare earth element but is suspected to be related to Alzheimer’s disease [20]. In this study, we used ‘Original ZM10’, free of rare earth elements and aluminum, and it is appropriate for medical applications with higher biocompatibility.

In our previous studies, both surface coating with biodegradable polymers and chemical treatment with hydrofluoric acid (HF) have been reported to be effective strategies for improving the corrosion resistance of Mg alloy stents [21,22]. Surface coating with biodegradable polymers is used not only for corrosion resistance, but also for loading drugs to inhibit in-stent restenosis [23]. HF treatment is an effective approach for improving the corrosion resistance of Mg alloys. An HF-treated Mg alloy stent can support the vessel wall in a coronary artery for several days, but it is not possible for a gradually degrading stent to maintain its radial strength for several months [22]. Therefore, another approach is required to improve the durability of Mg alloy stents.

It has been reported that poly(chloro-para-xylynene) (parylene C) coating has the potential to reduce the surface wettability of Mg alloy owing to its dense structure and hydrophobic surface [24]. Parylene C is widely used for various medical devices, such as conventional stents and catheters [25,26], because of its durability and biocompatibility [27]. It can also be formed on the surface of various biomaterials by chemical vapor deposition (CVD) [28].

However, surface coating of a bioresorbable Mg alloy stent with parylene C has not yet been reported. As stated above, HF treatment was effective for improving the corrosion resistance of a Mg alloy stent. The objective of this study was to examine the effect of the parylene C coating on the corrosion resistance of Mg alloy stent, particularly to evaluate if it could be resistant to shape changes caused by crimping on the balloon catheter and subsequent expansion.

## 2. Materials and Methods

### 2.1. Chemicals

Hydrofluoric acid (HF, 46% *w*/*w* aqueous solution), tetrahydrofuran (THF), and Eagle’s minimal essential medium (E-MEM) were purchased from Wako Pure Chemical Industries Ltd. (Osaka, Japan). Fetal bovine serum (FBS) was purchased from Cosmo Bio Co., LTD. (Tokyo, Japan). Poly(D,L-lactide) (PDLLA) and poly(ε-caprolactone) (PCL) were purchased from LACTEL Absorbable Polymers (Birmingham, AL, USA). Mg alloy ‘Original ZM10’ was developed and manufactured by Fuji Light Metal Co., Ltd. The component was analyzed with an inductively coupled plasma optical emission spectrometry (ICP-OES Agilent 720, Agilent Technologies, Santa Clara, CA, USA) as shown in Appendix A.

### 2.2. Stents

The Mg alloy stent made of ‘Original ZM10’ was designed, fabricated, and mirror-polished by Fuji Light Metal Co., Ltd. (Kumamoto, Japan) and Japan Medical Device Technology Co., Ltd. (Kumamoto, Japan). The specifications of the stent used in this study are shown in Appendix A. The structure of our original stent is shown in Appendix A.

The bare Mg alloy stents were immersed in 46% (*w*/*w*) HF aqueous solution for 24 h at room temperature. Then, they were rinsed with deionized water and acetone in turn, and we finally dried the samples at 55 °C under vacuum [22].

### 2.3. Parylene C Coating

The parylene C coating of the HF-treated Mg alloy stent was performed by Specialty Coating Systems coating service (Specialty Coating Systems Inc., Indianapolis, IN, USA) [24]. The thickness of the parylene C coating layer was measured by reflection spectroscopy using an optical thickness meter (OPTM-F2, Otsuka Electronics Co., Ltd., Osaka, Japan).

### 2.4. Polymer Coating

Additional polymer coatings were applied after the parylene C coating. First, the stent was coated with a soft polymer, poly(ε-caprolactone) (PCL), as a base layer, and poly(D,L-lactic acid) (PDLLA) with sirolimus (SRL; a conventional antiproliferation drug) as a top layer, as described in our previous report [21]. Briefly, for the base layer, PCL was dissolved in THF to obtain a 0.5 wt% solution. Then, 400 ± 30 µg of polymer was applied to each stent at 0.02 mL/min using an ultrasonic spray coater, Exacta Coat Ultrasonic Spraying System (Milton, NY, USA). For the top layer, PDLLA and SRL were dissolved in THF to obtain a 0.5 wt% and 5 mg/mL solution, respectively. Then, 250 ± 30 µg and 1.0 μg/mm^2^ of polymer and SRL, respectively, were applied to each stent at 0.02 mL/min using the ultrasonic spray coater.

### 2.5. Contraction and Expansion of the Stent

We used a manual crimp tool (13JSK, Blockwise Engineering, Chicago, IL, USA) to crimp the coating stent on a balloon-tipped catheter (Vega SDD, Arthesys, Saclay, France) to 1.2 mm in outer diameter, as described in our previous studies [21,22]. Then, before the expansion, the crimped stent was immersed in 37 °C cell culture medium (E-MEM) containing 10% fetal bovine serum (FBS) for 2 min to simulate the clinical stent implantation process. For the expansion of the stent, we inflated the catheter with an Encore™ 26 Inflator (Boston Scientific, Marlborough, MA, USA) and expanded the stents to 3 mm in outer diameter. 

### 2.6. Surface Observation

The surface morphology of the Mg alloy stents before and after balloon expansion were observed by field emission scanning electron microscopy (FE-SEM, JSM-7100F, JEOL, Tokyo, Japan) with a hot electron gun. The observe condition was at an accelerating voltage of 15 kV and a working distance of 15 mm by a secondary electron (SE) detector.

### 2.7. Evaluation of Corrosion Behavior

After expansion of the polymer-coated Mg alloy stents, the stents were immersed in 10 mL cell culture medium (E-MEM + 10% FBS) in an incubator with a condition of 5% CO_2_, 37 °C under 100 rpm agitation for 12 months. The supernatant medium of 100 µL was collected after immersion for 1 month and 6 months, using a Magnesium B-test Wako test kit (Wako Pure Chemical Industries Ltd., Osaka, Japan) to measure the amount of Mg ions released from the Mg alloy stent, following the manufacturer’s protocol [29]. Mean values were calculated from measurements for five different stents. After the incubation, the stents were removed from the medium and dried in a vacuum overnight.

Elemental analysis of the cross-sections of the stents after incubation for 1, 6, and 12 months was carried out using an X-ray energy dispersive spectroscopy (EDS) detector installed into the FE-SEM (JSM-7100F, JEOL, Tokyo, Japan). The EDS mappings were recorded with a JEOL resolution silicon drift detector and indicated the location of elements analyzed.

### 2.8. SRL Elution from the Polymer Layer

After the expansion of the stent coated with SRL-loaded polymer using the same method as described in Section 2.5, the stent was then immersed in 10 mL PBS in an incubator at 37 °C in the dark under 100 rpm agitation for 64 days. To evaluate the SRL release from the polymer, 1 mL supernatant solution after immersion for 1, 3, 7, 14, 21, 28, and 64 days was measured with an ultraviolet/visible spectrometer at 278 nm. The amount of SRL eluting was calculated from three different samples. The SRL elution rate was calculated based on the SRL loading density of each stent.

## 3. Results and Discussion

### 3.1. Parylene C Coating of Mg Alloy Stents

The thickness of the parylene C coating layer was approximately 0.5 ± 0.15 µm (data not shown). Normally, a coronary stent is made of flexible tube-like meshes that is crimped onto a balloon-tipped catheter, and then the balloon is deflated and the stent is expanded at a narrowed coronary artery [30]. Considering the changes in the surface structure of the balloon-expanded stent is therefore essential. The surfaces of the Mg alloy stents were observed using a scanning electron microscope (SEM). The bare Mg alloy stent had a smooth surface both before and after the balloon expansion (Figure 1a). The HF-treated Mg alloy stent had a smooth surface before the balloon expansion but showed small wrinkles after the expansion, with no large cracks (Figure 1b). Following parylene C coating, the stent surface was smooth before the expansion, while after the expansion small wrinkles, similar to those shown by the HF-treated Mg alloy stent, were observed, with no large cracks. Because the parylene C coating was very thin (~0.5 µm) and flexible, it was able to expand and follow the structural changes of the HF-treated stent without peeling off (Figure 1c).

### 3.2. Corrosion Behavior of the Parylene-Coated Mg Alloy Stents after Expansion

The corrosion behavior of the stents after the balloon expansion was examined over a 1-month and 6-month immersion in cell culture medium at 37 °C (Figure 2 and Appendix A). Mg ion release (%) represents the relative corrosion rate of Mg alloy stent calculated from the total amount of Mg alloy stent [29]. After 1-month immersion, the bare Mg alloy stent indicated a rapid corrosion rate calculated as 38%, while the HF-treated Mg alloy stent exhibited a corrosion rate of 13%. The HF treatment of Mg alloy stent enhanced the corrosion resistance due to a protective layer of MgF_2_ and Mg(OH)_2_ formed on the Mg alloy stent surface, which prevented rapid corrosion [22,23]. Additional coating of the HF-treated Mg alloy stents with parylene C resulted in complete resistance (almost 0% corrosion). To further evaluate the corrosion resistance of the parylene C coating on HF-treated Mg alloy stent, it was immersed in cell culture medium for 6 months (Figure 2B). Compared with HF-treated Mg alloy stents, parylene coating still exhibited excellent corrosion resistance over a 6-month period. The HF-treated Mg alloy stents showed the corrosion rate of 78% was due to the MgF_2_ content with high mechanical strength inducing cracks on the surface after balloon expansion [22,31]. Additional coating with parylene C perfectly improved this disadvantage. Kandala et al. also studied the corrosion behavior of Mg alloy (AZ31) stents with parylene C coating (without HF treatment) in the cell culture medium for 3 days [32]. Compared with bare Mg alloy stent, parylene C-coated Mg alloy stent improved the corrosion rate from 0.45 mm/year to 0.12 mm/year. However, the balloon expansion of parylene C-coated stents accelerated the corrosion of Mg alloy stents from 0.05 mm/year to 0.15 mm/year due to defects and openings in the parylene C coating [33]. Therefore, tight adhesion between MgF_2_ and the parylene C layer was thought to form a strong and stable water barrier that was not affected by the balloon expansion of the stent struts. 

Drugs to suppress restenosis are generally introduced in additional polymer layers; therefore, we examined the effect of subsequent PCL and PDLLA polymer coatings on the performance of parylene C in protecting the HF-treated Mg alloy stent from corrosion. In our previous study, coating with PCL and PDLLA was found to be the best approach for enhancing the corrosion resistance of the Mg alloy stent and achieving long-lasting drug release from the polymer layer [21]. The polymer-coated HF-treated Mg alloy stents showed similar corrosion resistance to that of the HF-treated Mg alloy stents without polymer coating. The polymer-coated, HF-treated, and parylene C-coated stent showed almost complete resistance to corrosion. This indicates that the additional polymer coating did not affect the complete corrosion resistance provided by HF treatment and parylene C-coating over a 1-month period. 

We examined further corrosion of the polymer and parylene C-coated HF-treated Mg alloy stent after 1-, 6-, and 12-month incubations by SEM images and EDS mapping of their cross-sections (Figure 3). At 1 month, the SEM images and Mg signal from the EDS mapping showed no significant degradation of the Mg alloy. At this stage, F and O signals were observed on the surface of the strut and corresponded to the HF treated layer and the polymer layer, respectively. After 6 and 12 months, the area of the Mg signal had decreased, and a small amount of the O signal was observed on the Mg signal. The O signal that emerged is attributed to corrosion products of the Mg alloy. P and Ca signals also indicated corrosion products, corresponding to magnesium phosphate and calcium phosphate [34]. The source of phosphate and calcium was the incubation medium (E-MEM + 10% FBS) for the corrosion resistance experiment. According to the EDS mapping, the shallow corrosion on the Mg stent surface under the coating layer was observed, even after immersion for 12 months. In addition, it was confirmed that the coating layer remained on the Mg alloy stents because of a slow corrosion rate after the 12-month incubation. 

### 3.3. SRL Release from the Polymer Layer

For clinical application of the cardiovascular stent, drug release from the polymer outer layer is key for suppressing restenosis after implantation. We therefore examined the release of the drug SRL from the polymer layer (Figure 4). The polymer layer had eluted approximately 60% of the SRL at 14 days, 70% at 28 days, and 90% at 64 days. Cypher—a drug-eluting stent made of stainless steel coated with parylene C, polyethylene-co-vinyl acetate, and poly-*n*-butyl methacrylate—eluted 80% of the loaded SRL over 1 month [35]. Ultimaster—a drug-eluting stent made of Co-Cr alloy coated with PDLLA and poly(L-lactide–co-ε-caprolactone)—showed 90% elution at 3 months [36]. The elution profile of the reported stent is therefore considered to be similar to those of Cypher and Ultimaster.

## 4. Conclusions

This study demonstrated that parylene C coating is an efficient method to improve the corrosion resistance of Mg alloy stent. According to a long-term evaluation of corrosion behavior, the parylene C coating is expected to be a promising surface coating of Mg alloy stents for clinical applications. The following conclusions are obtained:(1)Parylene C coating on a HF-treated Mg alloy stent exhibited a smooth surface without cracks after balloon expansion. Because the parylene C coating was very thin (~0.5 µm) and flexible, it was able to expand and follow the structural changes of the HF-treated Mg stent without peeling off.(2)Combining the parylene C coating with the HF treatment of the Mg alloy stent showed complete corrosion resistance for at least 1 month in physiological conditions. The tight adhesion between the MgF_2_ and parylene C layers was thought to form a strong and stable water barrier even when the structure of the struts was deformed.(3)The parylene C coating did not affect subsequent layers of biodegradable polymer coating applied to a long-lasting release of antiproliferative drug for 64 days to suppress restenosis.

## Figures and Tables

**Figure 1 materials-15-03132-f001:**
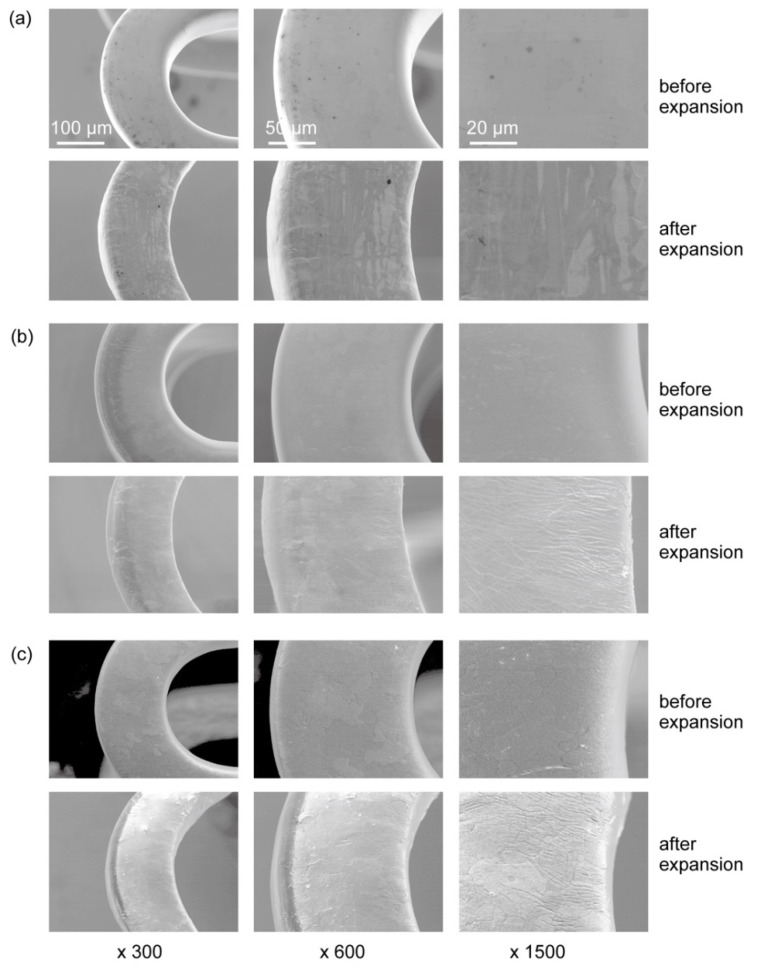
Surface morphology of original Mg alloy stents observed by SEM (**a**), HF-treated stents (**b**), and HF-treated Mg alloy stents coated with parylene C (**c**) before and after balloon expansion at different magnifications.

**Figure 2 materials-15-03132-f002:**
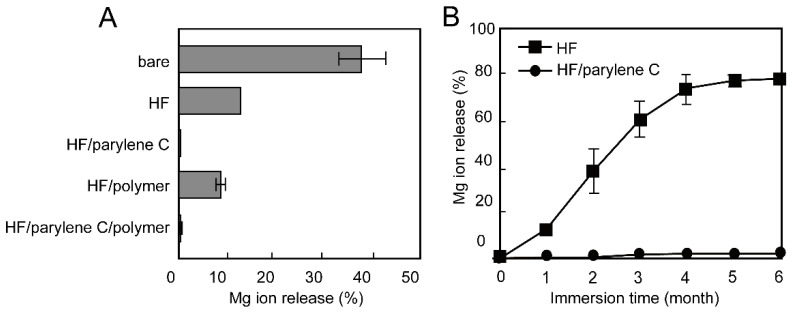
Mg ion release rate from Mg alloy stents with balloon expansion after 1-month (**A**) and 6-month (**B**) incubation in cell culture medium. The data represent the mean value for n = 5 and the error bars show the standard deviations of the means.

**Figure 3 materials-15-03132-f003:**
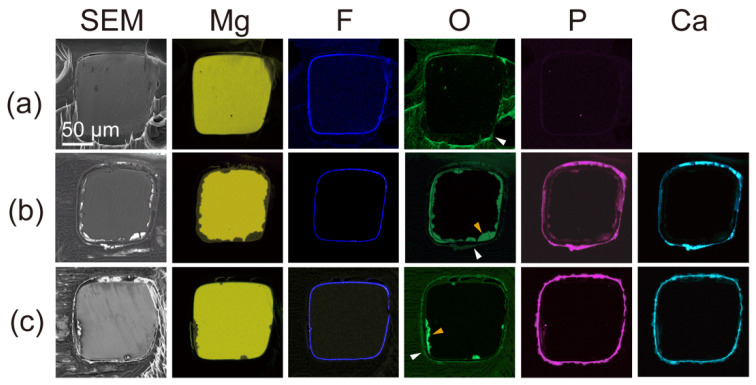
EDS mapping for cross-sections of parylene C and polymer-coated HF-treated stents immersed for 1 month (**a**), 6 months (**b**), and 12 months (**c**) in the cell culture medium. The white and orange arrow heads indicate O signal from polymer layer and corrosion products of the Mg alloy, respectively.

**Figure 4 materials-15-03132-f004:**
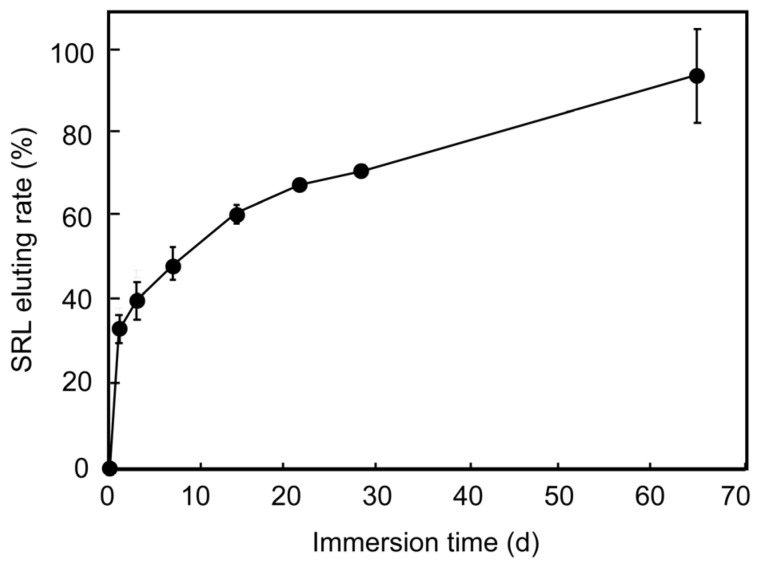
SRL-release profiles from the parylene C- and polymer-coated HF-treated stents after balloon expansion, in 37 °C PBS over 64 days. The data represent the mean value for n = 3 and error bars show the standard deviations of the means.

## Data Availability

Not applicable.

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
