# Peer review of "Effect of Parylene C on the Corrosion Resistance of Bioresorbable Cardiovascular Stents Made of Magnesium Alloy ‘Original ZM10’"

_materials, 2022, doi:10.3390/ma15093132_

Round 1

Reviewer 1 Report

Article: materials-1666745-peer-review-v1

Title: Effect of parylene C on the corrosion resistance of bioresorbable cardiovascular stents made of magnesium alloy.

  1. The title and the abstract should indicate the Mg alloy used in this study.
  2. The environment for corrosion testing should also be specified in the abstract.
  3. The nature of the polymers applied should also be indicated in the abstract.
  4. Line 149. Optical microscopy? Please revise. Figure 1 shows SEM micrographs.
  5. The quality of the micrographs in Figure 1 can be improved. The brightness/contrast is too low.
  6. Explain how the “Mg ion release (%)” was calculated. Does 100% correspond to the total amount of Mg ions that can be released from the stent? How precise is this calculation? The authors are encouraged to provide corrosion rate values in common units (mg cm-2 d-1).
  7. Why bother to include AZ31 and WE43 in Table S1? The composition of these alloys is not relevant for the current study. Please include Fe concentration in Table S1.
  8. Please indicate the manufacturing method of the ZM10 alloy (e.g. as cast, wrought).
  9. Lines 143 and 144 are out of place. This information should be in the experimental section.
  10. Please compare the results with those from other researchers.
  11. Higher magnification micrographs (cross-sections) should be included in order to assess the morphology and adhesion of the coatings.

Reviewer 2 Report

The article : Effect of Parylene C on the Corrosion Resistance of Bioresorbable Cardiovascular Stents Made of Magnesium Alloy present some interesting results for medical field. Few aspects must be taken in consideration in order to improve the material quality. 

In the Abstract section the second part must be re-phrased and better explaned (when was the polymer layer applied, for example). 

L38: a reference is required 

L40: are these Mg -alloy biodegradable and bioresorbable ? provide a study (reference) to confirm this statement 

L79: give a reference for chemical composition or mention the technique used for analysis 

L129: mention for SEM experiments the working tension of the filament (do you use low or high vacuum, metallic cover or free ?) , mention the EDS detector type and the analyze mode (mapping, line etc.) 

L146: provide a reference title 

L149: optical should be replaced with scanning electron ..... microscopy 

a better quality for all images from Figure 1 is necessary 

L164: how do you determine the corrosion rate ? (mass variation or thickness? ) 

Linear and cyclic potentiometry among EIS  can be performed in order to evaluate the role of the Parylene C layer and to confirm the conclusions. 

In figure 2 give the value of the standard deviation (±)

 The paragraph between L186 and L196 , XRD, XPS or AES results are necessary or references to confirm the compounds nature validity. 

Re-structure the conclusions is necessary 

Reviewer 3 Report

Although the proposed manuscript is interesting, there are enough weaknesses that need to be improved. This based on the following:

  • Line 17-25: I consider that the abstract should be reviewed again because it is very general and the objective is not clear.
  • The scope of the study is not well defined, the authors could better express it in the abstract
  • Line 29-70: The introduction must be enriched because the authors only place paragraphs that have no continuity. The authors should talk about the coatings, what is new, the types of magnesium alloys and of course the applications.
  • In which section of the article is the objective indicated, because it is not clear.
  • Line 72: Are the chemical substances used reported as a percentage by volume or by weight?
  • Line 116: In section 2.6, the authors must indicate the parameters used in the characterization by SEM (for example: operating at 20 kV and 12 and 8.5 mm working distance.), they must also say what the acronym SEM (scanning electron microscope) means.
  • What type of detector did they use: backscattered electron (BSE) and secondary electron (SE) detectors.
  • Line 162-170: The corrosion results are poorly stated, because the authors indicate deterioration percentage but that is not corrosion rate, they must make those changes. the corrosion rate is reported for example in mpy or in mm/year
  • Line 159: Figure 1 should be: SEM surface morphology .......
  • Line 188: It is important to indicate what EDX or EDS means?
  • Figure 3. It is necessary to indicate the percentage of elemental analysis
  • The discussion of results should be enriched, only the authors describe the results but they are not discussed.
  • Conclusions should be presented specifically and not as a paragraph.
  • it is necessary to indicate the repeatability of the tests
  • The authors present 23 references. There is no self-plagiarism, the authors must include more current references from the year 2020 and 2021.

Round 2

Reviewer 2 Report

Publish in the current form 

Author Response

Thank you very much for reviewing our revised-manuscript and for your very kind comment to us. 

Reviewer 3 Report

The authors have made all the suggested corrections, for which the article can be accepted for publication.

Line 85. it is wrong to put (HF, 46% w/w)

Line 130. …….15 mm by a secondary electron (SE) detector

The subsections of figure 1. must be (a), (b) and (c) in the images and in the caption

The subsections of figure 3. must be (a), (b) and (c) in the images and in the caption

Conclusions should be presented specifically and not as a paragraph.
